# Current Progress in the Utilization of Soy-Based Emulsifiers in Food Applications—A Review

**DOI:** 10.3390/foods10061354

**Published:** 2021-06-13

**Authors:** Lingli Deng

**Affiliations:** College of Biological Science and Technology, Hubei Key Laboratory of Biological Resources Protection and Utilization, Key Laboratory of Green Manufacturing of Super-Light Elastomer Materials of State Ethnic Affairs Commission, Hubei Minzu University, Enshi 445000, China; lingli0312@gmail.com; Tel.: +86-0718-8438-945

**Keywords:** emulsions, soy protein, soy polysaccharide, soy lecithin, delivery

## Abstract

Soy-based emulsifiers are currently extensively studied and applied in the food industry. They are employed for food emulsion stabilization due to their ability to absorb at the oil–water interface. In this review, the emulsifying properties and the destabilization mechanisms of food emulsions were briefly introduced. Herein, the effect of the modification process on the emulsifying characteristics of soy protein and the formation of soy protein–polysaccharides for improved stability of emulsions were discussed. Furthermore, the relationship between the structural and emulsifying properties of soy polysaccharides and soy lecithin and their combined effect on the protein stabilized emulsion were reviewed. Due to the unique emulsifying properties, soy-based emulsifiers have found several applications in bioactive and nutrient delivery, fat replacer, and plant-based creamer in the food industry. Finally, the future trends of the research on soy-based emulsifiers were proposed.

## 1. Introduction

The colloidal mixture of two immiscible liquids gives rise to an emulsion where one liquid is dispersed in another liquid. The solvent or substance that tends to form droplets in the emulsion is defined as the dispersed, while the liquid that surrounds the droplet is defined as the external or continuous phase. An emulsifying agent is a surface-active molecule with the ability to be adsorbed at the oil–water interface of the newly formed emulsion during the emulsification and protects the droplets from immediate coalescence. A surface-active molecule can be considered as a proper emulsifier if it possesses the ability towards frequent adsorption at the oil–water or air–water interface and causes a decrease in interfacial tension. There are low-molecular weight emulsifiers (e.g., soy lecithin) and high-molecular weight emulsifiers (e.g., soy protein and polysaccharides) in soy components. Low molecular weight emulsifiers decrease the surface or interfacial tension to a greater extent than the proteins or polysaccharides due to the partitioning of the entire molecule between the two phases. However, the proteins and polysaccharides would undergo a conformational change at the interface and there might be no clear definition of hydrophilic or hydrophobic groups, resulting in relatively high surface tension compared to that of the low molecular emulsifiers. Although low molecular weight emulsifiers are more effective in emulsion formation, the high molecular weight emulsifiers are more effective in the formation of the viscoelastic film around the oil droplets, which favors the stabilization of the food emulsions. In 2019, the world soybean production was 336.5 million metric tons, which accounted for approximately 58.9% of the world’s oilseed production. Soybeans contain about 40% protein and 20% oil and are considered to be a major source of proteins and oils, accompanied by the production of natural surfactants: soy protein, soy lecithin, and soy polysaccharides. Soy protein is the most popular plant protein source to serve as an ingredient in food formulation. It should be noticed that the functional properties of soy-based emulsifiers might be different among the ingredients from different suppliers. However, the functional properties (e.g., solubility) of commercial soy-based emulsifiers depend on the extraction and processing method utilized for its preparation.

## 2. Emulsifying Properties

Two aspects of emulsifiers are commonly studied in literature during the consideration of their emulsifying potential. These aspects include emulsifying activity and stability. Basic information regarding their emulsifying potential can be acquired by the determination of their efficacy to reduce the interfacial tension of emulsion [1]. Typically, interfacial tension is measured as a function of the increasing level of emulsifier and then calculated the surface tension versus concentration profile of emulsifier. However, the electrical properties of emulsifiers may have an impact on the stability and performance of an emulsion. Although various methods are available to measure the electrical properties of emulsifiers, micro-electrophoresis is considered the simplest and most widely used method. The important aspect of emulsifying properties includes their emulsifying activity, which can be assessed by two comparatively indexes: emulsifying capacities and emulsifying activity index. The emulsifying capacity of an emulsifier is described as the maximum oil content that can be emulsified in an aqueous medium containing a specific amount of the emulsifier without reverting or breaking down the emulsion, while the emulsifying activity index is described as the amount of oil that can be emulsified per unit emulsifier. 

Various analytical techniques and protocols are present for the investigation of emulsion stability in different aspects [2]. Among all, the emulsion stability index (ESI) established by Pearce and Kinsella [3], is a widely used and simplest method based on the analysis of the turbidity of the diluted emulsion at a specific wavelength. The phenomena affecting the emulsion stability mainly include coalescence, flocculation, creaming, sedimentation, Ostwald ripening, and phase inversion (Figure 1). Density-driven gravitation separation is the most easily observed emulsion instability phenomena in food emulsion storage, including the upward movement induced creaming and the downward movement induced sedimentation. Coalescence is the merging process of small emulsion droplets into larger emulsion droplets, resulting in the formation of a distinct layer of oil on the surface (oiling-off). Flocculation phenomena are also derived from the aggregation of emulsion droplets, while the droplets would not merge but remain as individual droplets. Phase inversion is a process that the dispersed phase becomes the continuous phase and vice versa, which has also been applied as an emulsification method to make fine emulsions. Ostwald ripening is the growth of one emulsion droplet at the expense of the smaller one due to the difference in chemical potential of the emulsion droplets. The destabilization of the soy-based stabilized emulsions is usually affected by the nature of the molecules (three-dimensional structure, hydrophobicity, charge, solubility, molecular flexibility) and conditions of the environment (e.g., pH and ionic strength) [4].

## 3. Soy Proteins

Soy proteins are mainly composed of 40% 7S (β-conglycinin) and 30% 11S (glycinin) fractions. The 7S globulin has been shown to have better emulsifying properties than the 11S globulin, which is due to the 11S globulin’s more stable oligomeric form. Rivas and Sherman [6] found that 7S formed a stronger viscoelastic film at the oil–water interface than 11S at all the tested pH and NaCl concentrations. They concluded that the 7S globulin molecules have a higher degree of intra- and intermolecular cohesion, resulting in more organized films at the interface. Glycinin cannot adsorb quickly to the air–water interface due to its low surface hydrophobicity, large molecular size, and low molecular flexibility. Liu et al. [7] observed that the acidic subunits of glycinin adsorb more quickly to the air–water interface than raw glycinin. Soy protein acts as a macromolecular surfactant to stabilize oil-in-water emulsion systems in food products such as sausages, ice cream, yogurt, and coffee whitener [8]. It has been proposed that the surface hydrophobicity and solubility are the major factors determining emulsifying activity, while the molecular flexibility of the proteins is important for emulsion stability [9]. However, the solubility of soy proteins was limited in the food matrix due to the isoelectric point being around 5.0. Moreover, the globular structure of the soy proteins prevents the exposure of the hydrophobic residues and retards the conformational change of protein when adsorbing at the interface, preventing them from becoming more efficient at reducing interfacial tension [10]. In recent years, soy proteins are considered effective starting materials to produce functional emulsifiers with modified physico-chemical properties by using different enzymatic, physical, chemical, and biological methods [11].

### 3.1. Enzyme Treatment

Enzyme treatment has been an effective method to modify the functional properties of proteins due to the advantages of high specificity and tunable properties of the hydrolyzed peptides. Enzymatic hydrolysis could reduce the molecular weight and expose the groups (hydrophobic or hydrophilic) buried in the globular structure of the soy protein, which might elevate the emulsifying properties. The tunable functional properties make the hydrolyzed soy proteins important ingredients in the food industry [10]. In a recent report, three different enzymes were reported namely flavorzyme, alcalase, and protamex to hydrolyze SPI to obtain hydrolysate with degrees of hydrolysis of 3%, 7%, and 11% [12]. Flavorzyme was found to induce soy protein nanoparticles (SPNPs) at all DH, whereas protamex showed a limited effect on the formation of SPNPs. The SPNPs attached more rapidly at the oil–water interface as compared to native SPI, indicating the good emulsifying potential of the emulsifier. The peptides with decreased molecules show the advantage of higher mobility when absorbing at the interface, which might also be a disadvantage in maintaining the emulsion stability. Therefore, even though most of the previous studies indicated that that enzymatic hydrolysis improved some of the functional properties of soy proteins, the degree of hydrolysis is the critical parameter for enzyme modification [13]. Enzyme treatment could not only hydrolyze the protein, but also crosslink protein to modify the emulsifying properties. Zang et al. [14] hydrolyzed soy protein by papain with a hydrolysis degree of 6% and then reacted with transglutaminase (TGase) to partial crosslink soy protein hydrolysate. Compared with the raw SPI, transglutaminase treated SPI and SPI hydrolysate, emulsion stabilized by transglutaminase treated SPI hydrolysate exhibited the lowest particle size, creaming index, flocculation degree, coalescence degree, and highest freeze–thaw stability.

### 3.2. Thermal Treatment

The conformation and aggregation behavior of proteins are greatly influenced by heat treatment, resulting in modified functional properties [15]. The protein molecules tend to unfold the globular structure and expose the hydrophobic groups. It has been reported that the emulsions stabilized by 75 °C; heated soy protein showed lower emulsion droplet size compare to the unheated soy protein. The heated soy protein showed a higher extent of adsorption at the interface, with a higher ratio of β-conglycinin among the absorbed protein. The acidic and basic subunits of glycinin remained in the serum phase in similar amounts in these emulsions as in the unheated samples. Heat-induced dissociation of β-conglycinin has previously been shown to result in the formation of soluble complexes [16,17]. However, high-temperature treatment (95 °C for 15 min) might cause dissociation and denaturation of both β-conglycinin and glycinin, resulting in the formation of soluble aggregates between the basic subunit of glycinin and the β-subunits of β-conglycinin. The majority of the aggregates formed during heating adsorb at the interface during homogenization, which could explain the decreased emulsion stability [15].

### 3.3. Non-Thermal Processing

Non-heat processing (such as ultrasonication and high pressures) has long been researched to modify the functional property of soy proteins. Ultrasound is well-known energy and time-saving technique. The use of ultrasound in various industries is becoming more common. Ultrasound destroys noncovalent interactions and disulfide bonds through thermal, mechanical, and chemical effects [18], thus causing protein subunits to dissociate and aggregate, resulting in the modification of solubility, emulsifying, foaming, and gelation properties [19]. Jambrak et al. [20] found that ultrasound treatment (20 kHz) increased the emulsifying and foaming ability of soy proteins, which might be ascribed to the higher adsorption at the emulsion droplet interface of the denatured soy protein [10]. It has been revealed that ultrasound treatment increased the surface hydrophobicity and zeta potential value of soy-protein isolate–citrus-pectin electrostatic complexes, and significantly decreased emulsion droplet size was observed. The results showed that ultrasonic cavitation effects changed the structure of both biomacromolecules and increased the electrostatic interactions between soy-protein isolate and citrus pectin, both of which led to the complex’s improved emulsifying properties [21]. Albano et al. [22] have also observed reduced particle size in the soy protein and pectin complex stabilized emulsion after ultrasonication. Ren et al. [23] compared the effects of ultrasonic cavitation treatment and hydrodynamic cavitation on the functional properties of soy protein isolate. The particle size and viscosity of SPI were reduced and the surface hydrophobicity was increased after ultrasonic cavitation or hydrodynamic cavitation treatment, resulting in improved solubility, emulsifying activity index, emulsion stability index, and foaming capacity. However, significantly decreased foam stability was observed after ultrasonic cavitation and hydrodynamic cavitation treatment, which might be ascribed to the weaker protein-protein interaction as reflected by the decreased viscosity.

When proteins are subjected to high pressures (HP), it is known that protein molecules would undergo conformational changes, which may lead to modification of the emulsifying properties. Several studies have been conducted to see whether high-pressure treatment can be applied to change the emulsifying properties of soy proteins. Molina, Papadopoulou [9] reported that the emulsifying activity was increased by high-pressure treatment at neutral, while no improvement was observed for the emulsifying stabilities and solubilities. They observed the highest emulsifying activity indexes were 400 MPa and 200 MPa for β-conglycinin and glycinin, respectively. Puppo et al. [24] compared the effect of high pressure on emulsifying properties of soy protein isolate at acidic (pH 3.0) and alkaline (pH 8.0) conditions. High-pressure treatment (200 MPa) of SPI at alkaline condition induced a reduction of droplet size and an increase of depletion flocculation. High-pressure treatment induced a significant increase of adsorbed proteins at the oil–water interface at both pH conditions. Torrezan et al. [25] found in both the low pH range (2.66–4.34) and the near-neutral range (5.16–6.84), increasing the soy protein concentration (0.32–3.68%) caused a reduction in emulsifying activity index. In acidic conditions, the emulsifying activity index was higher at low-pressure treatments whereas, in the near-neutral pH range, the best emulsifying activity was at the middle range of pressure treatment. The critical effect of the initial protein concentration was confirmed by Wang et al. [26], who observed higher EAI value when lower soy protein concentration (1%) was applied for high-pressure treatment. Apart from some common non-heating processes that have been applied for modification of the emulsifying properties of soy protein, pulsed electric fields (PEF) [27], extrusion process [28], and radiation [29] have also been studied.

### 3.4. Glycation Modification

Traditional chemical modifications of proteins, such as acylation, phosphorylation, and alkylation, are less researched in food applications due to safety and environmental concerns [30]. Protein non-enzyme glycation, commonly regarded as the initial stage of the Maillard reaction, has been widely researched in the food industry to modify the functional properties of protein due to the relatively mild and safe reaction conditions, and no extraneous chemicals were needed [31]. Thus, this makes glycation a promising method for protein modification in the food industry [32]. There have been emerging studies that focused on improving the stability of glycated soy protein hydrolysates stabilized emulsions. Recent studies that focused on the glycation of polysaccharides with soy proteins for improved emulsifying properties are listed in Table 1. Covalent attachment between proteins and polysaccharides may enhance the protein functionality to act as both emulsifier and stabilizer. Most of the investigations (Table 1) conducted on Maillard conjugates followed a similar trend that glycated conjugates increased the emulsification ability and emulsion stability. The main advantages of the soy protein-polysaccharide conjugates synthesized by the Maillard reaction include the increased functional characteristic and solubility over a wide range of environments, such as very low pH, and very high ionic strength [33]. In the case of larger molecular weight polysaccharides, the conjugate-stabilized emulsion may have a thicker stabilizing layer than the protein-stabilized ones (Figure 2). It has been reported that the well-prepared soy protein hydrolysate–polysaccharide conjugates substantially improve emulsifying and stabilizing properties as compared with soy protein hydrolysates and its native proteins [34]. Apart from polysaccharides, some low molecular weight carbohydrates such as glucose and maltose have also been reported to be conjugated with soy protein hydrolysates. Yang et al. [35] studied the effect of the chain length of the carbohydrate on the interfacial and structural characteristics of the conjugates of soy protein hydrolysates (Mw > 30 kDa) produced by the Maillard reaction. The results of the study revealed that increasing carbohydrate chain length increases the emulsion stability of the conjugates. Zhang et al. [36] reported that the soy protein hydrolysate–dextran conjugate-based emulsions exhibited better freeze–thaw stability compared with the SPI-dextran conjugates, especially in the case of 3% DH. For the lowest creaming index and best freeze–thaw stability, the optimum wet Maillard reaction conditions included a soy protein hydrolysate/dextran ratio of 2:3 in which the dispersion of 40 g/L of soy protein hydrolysate was prepared in phosphate buffer (pH 8), which was then incubated for 1 h at 85 °C. The surface activity measurements indicated the closely packed soy peptide–dextran conjugates which formed a thick adsorbed layer at the oil–water interface. Even though the Maillard reaction has long been regarded as an effective method to improve the emulsifying properties of soy protein [31], there are still concerns about the conformational change, surface polarity, specific interactions among the components. Due to the complexity of the Maillard reaction, the productivity, stability, and repeatability of the conjugates are concerns for commercial use.

### 3.5. Fermentation Modification

As an ancient processing approach, lactic fermentation has long been applied on soy milk or soy protein modification for improved sensory properties [44]. However, few studies focused on the effect of fermentation on the emulsifying properties of soy protein. The fermentation process would not only undergo enzymatic hydrolysis on soy proteins, but also affect the conformation of protein molecules through acidification, resulting in a comprehensive impact on the functional properties of the protein. Meinlschmidt et al. [45] studied the effect of liquid state *lactobacillus helveticus* fermentation on the solubility, emulsifying capacity, and foaming activity of soy protein isolate. The fermentation significantly decreased the SPI solubility at pH 7.0, while increased the solubility at pH 4.0. Non-fermented SPI exhibited the highest emulsifying capacity of 660 mL/g, while fermentation resulted in a significantly decreased emulsifying capacity (475–483 mL/g), which was ascribed to the decreased solubility at pH 7.0. The foaming activity of SPI was nearly doubled after fermentation, while the foam density decreased after fermentation.

It should be noted that the modification method is not limited to those discussed above. Indeed, each modification method has some advantages and disadvantages that affect the utilization in the food industry. For example, the physical modification has the advantage of high productivity, while usually showed low effectiveness on the emulsifying properties improvement. Enzymatic hydrolysis was widely recognized as a cost-effective way to modify the functionalities of proteins due to its controllability and minimal formation of by-products. However, the deeply hydrolyzed protein is generally accompanied by the production of bitter peptides, which has an adverse effect on the product flavor. Traditional chemical modification usually has safety and environmental concerns, especially when applied in the food matrix. The non-enzyme glycation is environmentally friendly, safe, and easily acceptable by consumers. The most recent studies on the modification of soy proteins for improved emulsifying properties focused on the combination of novel technologies with traditional modification methods. Wen et al. [46] applied a novel slit divergent ultrasound to facilitate the formation of soy protein isolate–lentinan conjugates via Maillard reaction. The results showed that ultrasonic treatment (40 min) markedly increased the degree of grafting (26. 48%) compared with the traditional heating method (2 h, 13.89%). The hydrophobicity, emulsifying activity, and emulsion stability were doubled after ultrasonic treatment for 40 min and the SPI-lentinan conjugates stabilized emulsions were stable against the various environmental stress (pH, temperature, and ionic strength).

## 4. Soy Polysaccharides

Soybean soluble polysaccharide (SSPS), a by-product of isolating soybean proteins, has been reported to be used as an emulsifier for the emulsification of beverages owing to the acidic nature of polysaccharide. Rhamnogalacturonan backbone is present in the SSPS structure, which is branched by β-1,4-galactan and homogalacturonan, α-1,3-, or α-1,5-arabinan chains [47] (Figure 3). It has been reported as a source of dietary fibers in fortified foods as well as a functional ingredient for food and pharmaceutical applications. The conformation of SSPS is not easily affected by pH and ionic strength which results in environmental stability of the SSPS stabilized emulsions. The structure of glycoprotein present in SSPS is almost similar to that of the Wattle Blossom Model suggested for gum arabic. The attachment of carbohydrate functionality of the polysaccharide on the oil–water interface is mainly due to the protein fraction of SSPS. Therefore, the hydrophilic portion of SSPS forms a 30 nm thick hydrated layer that retard the chance of coalescence and stabilizes the oil droplets by steric repulsion [48] (Figure 3).

Generally, the emulsifying ability of SSPS is affected by the protein fraction, molecular weight, and extraction conditions. Nakamura, Takahashi [47] studied the emulsifying potential of three different types of SSPS and observed that all the soy polysaccharide stabilized emulsions showed stability against creaming for 30 days at pH 3.5–5.0 when the polysaccharide concentration was above 4%. The soy polysaccharide that was extracted at pH 3.0 and 120 °C for 2 h showed the best emulsifying ability, in which the protein fraction had an inevitable effect. Nakamura et al. [49] separated the SSPS into two fractions, i.e., high molecular and low molecular, and found that the 2.2% protein facilitated the emulsifying properties of the high molecular fraction but did not exhibit a similar effect on the low molecular fraction of SSPS. The emulsions stabilized by the high molecular fraction showed no change when heated at 90 °C and pH 3.0–7.0 or in the presence of < 10 mM CaCl_2_, while the low molecular fraction stabilized emulsions undergo aggregation when heated at pH 7.0 [50]. The SPSS was further enzyme-digested by pectinases (polygalacturonase (PGase), hemicellulases (galactosidase (GPase), and rhamnogalacturonase (RGase)), and arabinosidase (Afase). The Rgase digested SSPS showed improved emulsifying properties while the others compromised the emulsifying potential [51]. The additions of SSPS to the protein-stabilized emulsions have also been reported to progress the stability against thermal treatment, low pH, and under simulated gastric conditions. Yin et al. [52] fabricated the stable nano-sized emulsions with the help of soy protein and SPSS complexes, which formed the interfacial films under the influence of the temperature by the process of electrostatic complexation of the denatured protein and soy polysaccharide. The interfacial fixed polysaccharide chains are also able to stabilize the oil droplets in an aqueous medium even in the unfavorable condition of soy protein in which they undergo aggregation.

Recently SPSS has shown potential applications in food emulsion products, such as beverages and mayonnaise. Nakamura et al. [53] used SSPS as a stabilizer in the dispersions of acid milk and suggested the comparable stabilizing potential of SSPS with pectin. It was also found that SSPS did not show interaction with casein at pH > 4.6, but exhibited better stabilizing ability at pH < 4.2 than high methoxyl pectin [54]. Chivero et al. [55] examined the ability of SSPS to produce O/W mayonnaise-like emulsions and observed that SSPS could stabilize emulsions with a maximum oil content of about 60 wt%, and the emulsions remained stable after 30 days. The improved stability was observed when SSPS was combined with a thickening agent (xanthan gum) to induce a stronger network. Xu et al. [56] fabricated casein and SSPS compact complex aggregates of 133 nm (Figure 4a), leading to the stabilized emulsions having the stability of more than 500 days with a curcumin loading efficiency of 99.9% and droplet diameter of about 324 nm (Figure 4b). It was also found that the absorption of curcumin was more effective compare with the absorption of the curcumin/Tween 20 suspension group, resulting in 11-fold higher oral bioavailability of curcumin in the emulsion group (Figure 4c). Zhan et al. [57] studied the SSPS effect on the functional characteristics of pea protein isolate (PPI), and found that SSPS adhered to PPI by means of hydrophobic interaction and hydrogen bonding which resulted in decreased hydrophobicity of the surface of reconstituted PPI particles and enhanced the stability of emulsions. It was also suggested that the incorporation of SSPS rearranged and interconnected the modified particles, resulting in the improved interfacial and rheological properties of the emulsions.

Apart from SSPS, the insoluble soy polysaccharide (ISP) containing cellulose, hemicellulose, lignin, and a protein fraction, has attracted attention due to its potential as a Pickering emulsion stabilizer. Porfiri et al. [58] performed the acidic extraction at pH 3.5, 120 °C and extracted insoluble soybean polysaccharide (ISPS) from insoluble okara. The pretreatments (high-pressure homogenization or sonication) are assumed to expose the internal site of the structure of protein and polysaccharide to enhance the superficial hydrophobicity. This in turn facilitates the formation of the outer layer and/or absorption of the macromolecules at the oil–water interface, hence providing increased rigidity of the interfacial film. Particularly, the molecules that resulted from high-pressure homogenization exhibited promising emulsifying potential and showed stability against the pH variation of the emulsions. Mwangi et al. [59] found that ISP dispersions under a high power ultrasonication treatment result in the breakdown of polysaccharide fibers and allow the preparation of the nanoparticles with a size range of 127–221 nm. The fabricated nanoparticles exhibited remarkable potential towards emulsification and allow the formation of Pickering emulsions. It was further reported Yang et al. [60] also fabricated ultrasound-induced insoluble soy polysaccharide nanoparticles with a size of about 160 nm, which can be served as an outstanding Pickering stabilizer for the emulsions having a high internal phase due to the formed gel network (Figure 5). The high internal phase Pickering emulsions showed high stability against environmental stress. The gel structure can be maintained over the pH range 2.0–12.0 and ionic strength range 0–500 mM. All the high internal phase Pickering emulsion gels exhibited excellent stability against prolonged storage and heating, as well as unique reversibility of freeze–thawing-destabilization/re-emulsification.

## 5. Soy Lecithin

Soy lecithin has been an important emulsifier for the production of food emulsion products. Commercial soy lecithin contains 34% triglycerides, 65–75% phospholipids, and small amounts of pigments, carbohydrates, sterol glycosides, and sterols. The commonly found phospholipids include phosphatidylcholine (29–46%), phosphatidylethanolamine (21–34%), and phosphatidylinositol (13–21%). Due to their amphiphilic nature, they can easily be adsorbed onto the surface or interface with the hydrophobic tail of fatty acid facing oil phase while the polar head group facing aqueous phase that results in decreased surface or interfacial tension. The phosphatidylinositol stabilizes the emulsion by serving as a barrier at the surface of oil or water droplets. The phosphatidylcholine and phosphatidylethanolamine contain the positively charged choline and ethanolamine groups, and the negatively charged phosphate and carbonyl groups. The soy phospholipids may form liposomes, micelles, lamellar structures, or bilayer sheets in an aqueous medium depending on the hydration, temperature, and concentration [10] (Figure 6). The described self-assembly systems are considered as potential delivery vehicles for bioactive and nutrients. Since commercial soy lecithin is a mixture of various phospholipids and other numerous constituents, its surface activity is a combined effect of all the surface-active substances. Though lecithin is not usually considered a suitable material to stabilize either oil-in-water or water-in-water emulsions, it can only be utilized for the preparation of emulsions at appropriate salt concentration, pH, temperature, and oil/water ratio.

Soy lecithin has been an effective emulsifier to fabricate delivery systems for enzymes, nutraceuticals, vitamins, flavors, pesticides, and antimicrobials. Flores-Andrade et al. [61] compared the emulsifying properties of soy lecithin and gum arabic for the fabrication of paprika oleoresin nanoemulsions and observed the remarkable efficacy of soy lecithin to form nano range droplets (d < 150 nm) rather than gum arabic (d < 539.6 nm). Yang et al. [62] prepared the lecithin stabilized emulsions with a droplet size range of 62.5–105 nm for co-delivery of essential oils and curcumin. It was found that the solubility of curcumin was significantly increased by 1700-fold, and its in vitro bioaccessibility was 4.79–10.6 fold higher than that of free curcumin. Koo et al. [63] prepared the emulsions stabilized by 1.5% soy lecithin, 0.5% sodium caseinate, and a combination of both (0.5% sodium caseinate and 0.5% lecithin). The sodium caseinate stabilized emulsions containing and the emulsion stabilized by the mixture of both (sodium caseinate and soy lecithin) undergo destabilization at pH 5 or below because of the aggregation of sodium caseinate near its isoelectric point. While the soy lecithin stabilized emulsion showed stability over a pH range of 3.5 to 7 because of increased repulsion among the droplets.

Soy lecithin has also been added to the protein stabilized emulsions to improve the emulsion stability through surfactant-protein interactions. García-Moreno et al. [64] investigated the influence of a combination of casein and phospholipid (0.3% and 0.5%, respectively) on the oxidative and physical characteristics of 10% fish oil-based emulsions at pH 7. Three different phospholipids were used to conduct the analysis which include lecithin, phosphatidylethanolamine, and phosphatidylcholine. The lecithin stabilized emulsion exhibited the best physical stability as they possess larger negative zeta potential with smaller droplet size. Additionally, they possess a smaller degree of oxidation which may be attributed to the combined effect of casein and lecithin, which results in favorable thickness and structure of the interfacial layer capable to prevent the oxidation of emulsion lipid. Wang et al. [65] added soy lecithin (0.5–1.0%) in the whey protein stabilized emulsions, and found improved emulsion stability because of the surfactant–protein interactions at the interface, resulting in a higher encapsulation efficiency of the spray-dried microcapsules with good re-dispersibility in water. Shen et al. [66] investigated the interactions of soybean lecithin with egg yolk granules, and they observed that incorporation of lecithin destroyed the aggregated structure of egg yolk granules, leading to better stability of emulsion because of lower particle size and higher surface charge.

Recently, a few studies have focused on constructing some novel soy lecithin-based emulsions with interesting functionalities. Sandoval-Cuellar et al. [67] fabricated the high oleic palm oil nanoemulsions, which were stabilized by the whey protein and soy lecithin and observed the less release of free fatty acid in in vitro gastrointestinal digestion as compared to non-encapsulated control. This is because of the lipase inhibition potential of soy lecithin. Zhuang et al. [68] inoculated *Bifidobacterium lactis* and *Lactobacillus acidophilus* into the lecithin-based oleogel emulsions prepared by using 20 wt% oleogelators (stearic acid: soy lecithin = 5:5), 70% canola oil, and 10% water. The semi-solid oleogel emulsions based on soy lecithin improved the viability of the encapsulated probiotics and prevented the progress of lipid oxidation. Jiang et al. [69] designed edible Pickering emulsions of high internal phase bearing a double-emulsion morphology (Figure 7). They dissolved the lecithin in squalane oil, and the dispersion of zein nanoparticles was prepared in water (w1). When the system allowed to emulsify for the first time, a w1/o emulsion was formed. The resulting primary w1/o emulsion was added with the second dispersion of zein nanoparticles (w2), which on emulsification leads to a w1/o/w2 double emulsion, the total ratio of oil and water was maintained at 3:1. It was found that soy lecithin enhances the surface elasticity of the interfacial films and resulted in highly stable emulsions.

## 6. Food Applications

### 6.1. Bioactive Encapsulation and Delivery

Recently the interest in the fabrication of novel bioactive delivery systems increased continuously [70]. In food applications, various emulsion-based delivery systems are used but they still required long-time physical stability in various environmental conditions because the breakdown of emulsion significantly alter the texture, color, flavor, and shelf life of the products. Xu, Wang [56] found that the curcumin oral bioavailability of the casein–soy soluble polysaccharide complex stabilized emulsions was 11-fold higher compared with curcumin/Tween 20 suspension. Wang et al. [71] found that the addition of soy polysaccharides (soy hull polysaccharide and soy soluble polysaccharide) is capable to decrease the influence of simulated gastric fluid (i.e., pepsin, ionic strength, and pH) on the stability of emulsions.

In recent years, a few novel emulsions such as Pickering emulsions [60], emulsion gels [72], oleogel emulsions [68], high internal phase emulsions, multiple emulsions [69,73], nanoemulsions, and microemulsions [74] have been fabricated for encapsulation and delivery of bioactive compounds (Figure 8). Physically stabilized Pickering emulsions with solid particles that were moderately wetted by oil and water showed improved stability against steric mechanism-based flocculation and coalescence [59]. Soy-based emulsifiers are considered promising Pickering stabilizers due to easy availability, the ability to form nano-aggregates, and health effects. Liu and Tang [75] reported the heat-treated soy glycinin stabilized gel-like Pickering emulsions capable of sustained release β-carotene which was confirmed by the in vitro experiment for intestinal digestion which indicated that the formation of a gel-like network significantly slowed down the release of β-carotene. Muñoz-González et al. [76] produced four emulsion gels containing soy protein, olive oil, and alginate-based cold gelling agent to encapsulate polyphenol. The emulsion gels with added polyphenols exhibited the presence of gallic acid, flavanol monomers, and their derivatives, which play a vital role to make it a suitable system for the delivery of various bioactive compounds. Zhuang, Gaudino [68] fabricated novel soy-lecithin based W/O oleogel emulsions for improved lipid stability and probiotic viability. The oleogel emulsion was composed of 20 wt% oleogelators (soy lecithin: stearic acid, 1:1), 70% canola oil, and 10% water. Flores-Andrade, Allende-Baltazar [61] compared the O/W paprika oleoresin nanoemulsions which were stabilized by whey protein concentrates, soy lecithin, and gum arabic under high-pressure homogenization, and found that soy lecithin was the most effective emulsifier for nanoemulsion preparation.

### 6.2. Fat Replacer

To mimic the properties of animal fat, soy-based emulsified systems have been researched as a promising alternative to replace animal fat [77,78]. Emulsion gel structures based on soy protein have the potentials to replace fat in sausages [79,80]. Pintado et al. [81] fabricated the soy protein stabilized emulsion gels that contained two different solid extracts of polyphenols obtained from grape seed and olive to be used as an animal fat replacer in the development of frankfurters.

The addition of SPI in ice creams may decrease the influence of heat on the recrystallization and melting rate of ice. Chen et al. [82] reported that larger-sized SPI nanoparticles with higher surface hydrophobicity and enhance the potential of packing at the oil–water interface are more suitable to form Pickering emulsions with enhanced freeze–thaw stability. The addition of soy protein in ice cream reduces the heat shock effect on the melting rate and recrystallization of ice. Functional optimization of the structure of soy protein is currently attracted great interest to achieve a high degree of fat globule partial coalescence in the preparation of ice cream. Soy protein under selective hydrolysis provides sufficient fat partial coalescence and good melt-down rates. Chen et al. [83] compared the effect of commercial soy protein isolate (CSPI), native soy protein isolate (NSPI), soy protein hydrolyzed by pepsin (SPHPe), and skim milk powder (SMP), and soy protein hydrolyzed by papain (SPHPa) on ice cream mix stability and melt-down properties. Among all these, the highest melt-down rates were observed in the ice cream made with SPHPe, which ranged from 1.23% min^−1^ to 2.05% min^−1^. While the CSPI, SMP, and SPHPa based ice creams exhibited the lowest melt rate with no significant difference from each other. The highest fraction of protein at the fat globules during aging and freezing was SPHPa mix (22.6%) which was almost similar to SMP (21.8%).

Currently, the soybean has been considered as popular among all the plant-based sources used for yogurt production, because of its quantity, quality, and functional characteristics of the protein. Such plant-based systems upon acidification cause destabilization of soy proteins, which may result in the formation of a non-continuous, weak gel [84]. SPNPs stabilized emulsions have been studied as functional non-dairy yogurts. Sengupta et al. [85] found that the addition of SPNPs distinctly improved the quality of soy yogurts with enhanced radical scavenging activity and ferric reducing property. The SPNPs (72.42–586.72 nm) incorporated soy yogurts exhibited significantly enhanced oxidative stability against peroxidation of lipids, indicating the applicability in the development of functional yogurts.

Consumers are purchasing plant-based “milks” and “creams” frequently because of their environmental and perceived health benefits. Hence, the food companies have been developing a wide range of plant-based milk and creamer products including soy products. Chung et al. [86] applied soy lecithin (1–5%) to stabilize 10% O/W emulsions, which remained physically stable upon addition of an acidic hot solution of coffee and retard the phase separation or increment in particle size. The soy lecithin stabilized oil droplets exhibited the enhanced surface negativity, hence they exert the strong electrostatic repulsion among the droplet and retard their aggregation. Koo, Chung [63] found the ability of soy lecithin to replace the caseinate in coffee creamers. The model O/W emulsions stabilized with a mixture of emulsifiers based on sodium caseinate (0.5%) and soy lecithin (0.5%) showed physical stability over a pH range of 5.5 to 7.

## 7. Challenges and Future Trends

It should be noted that the commercial soy-based products are changing along with the development of pro-cessing technologies. The functional properties of soy-based emulsifiers are sensitive to the processing history, such as enzymatic, thermal, and jet cooking processes [87,88,89], which makes the soy ingredients used in literatures varied. Hence, understanding the physiochemical states of various soy-based emulsifiers is necessary for better application of that in food products. Lee, Ryu [88] compared the solubility of commercial soy protein from different manufacturers and they found the soy protein isolates can be classified into three groups. One group had high solubility near the pl. Another group had low solubility near the pl, but high solubility at pH 11. The third group had low solubility even at pH 11. Recently, Zheng, Wang [89] selected 20 soy protein isolate samples from three manufacturers and they found that the best sample was Chiba tofu due to high hardness and springiness, and excellent quality. Hence, the researchers should identify their soy ingredients and select the most appropriate starting materials for their future academic or product development work.

Recently, various studies focused on the effect of novel technologies on the conformation, molecular interaction, and functional properties of soy proteins, soy polysaccharides, and soy lecithin, which provide fundamental understanding of the mechanism of their functional properties in various food processing conditions. However, there are still some challenges on the modification process on the soy-based materials. In future studies, the combination of the physical, chemical, enzyme, or biological methods to improve the emulsifying properties and other functional properties will be a trend for the development of novel food emulsifiers. However, the main challenge for the development of such novel emulsifiers is to select a stable fraction with high functionality under certain environmental conditions for the desirable formulated products. Therefore, more studies are needed to understand the relationship between the functional properties and molecular structure of these components before use in practice. The combination of soy based low molecular weight emulsifiers and high molecular emulsifiers (proteins and polysaccharides) will be a challenge in extending their applications to new fields.

## Figures and Tables

**Figure 1 foods-10-01354-f001:**
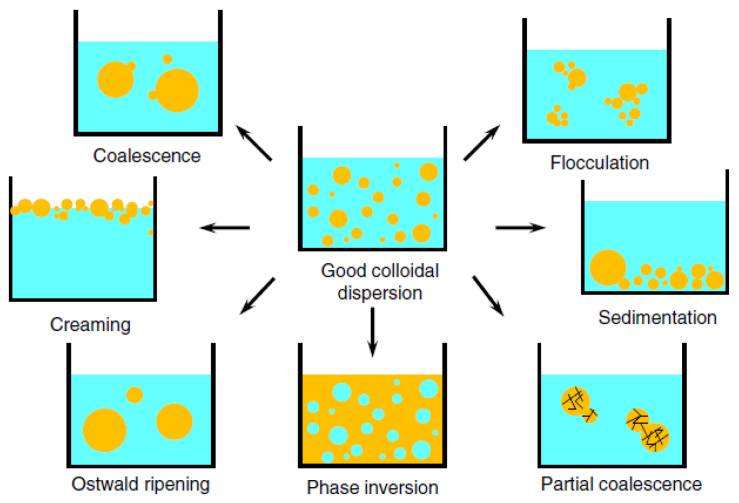
Mechanism for the destabilization of emulsion system [5].

**Figure 2 foods-10-01354-f002:**
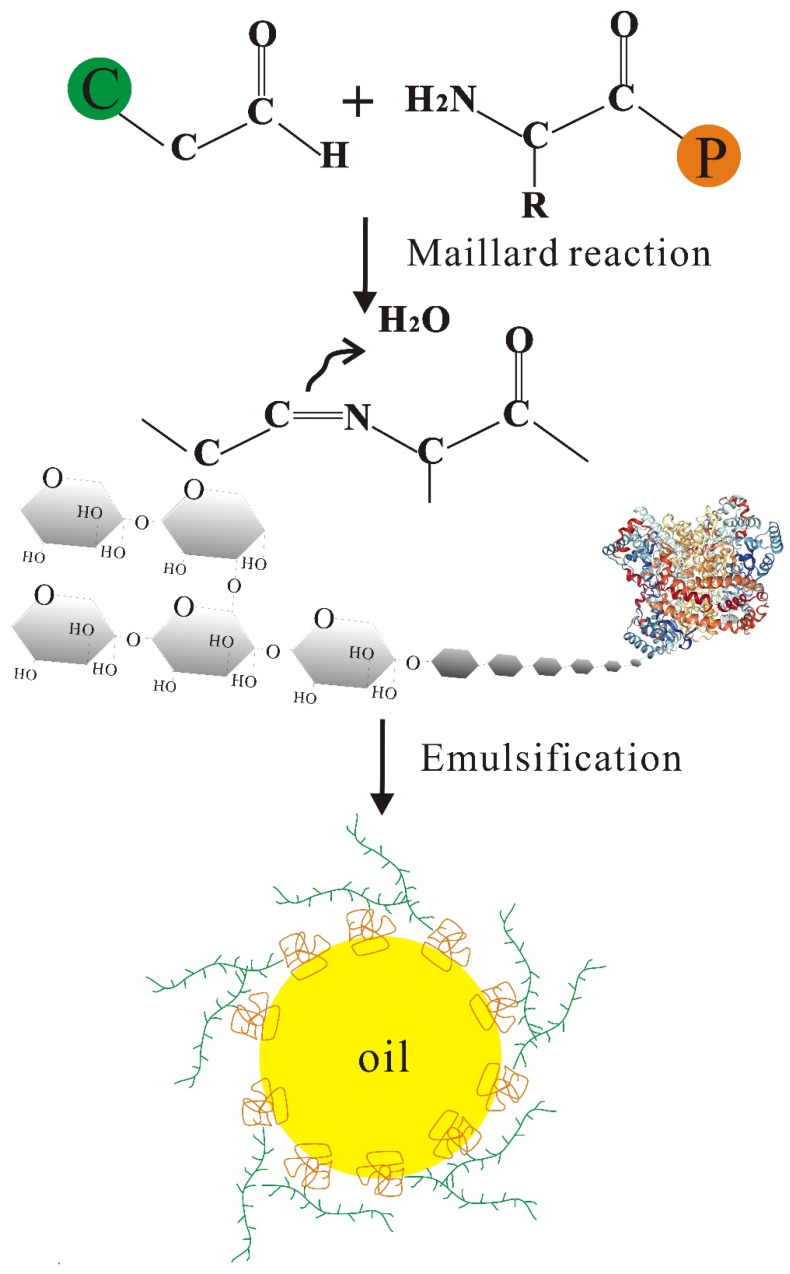
The schematic illustration of the formation of the soy protein–polysaccharide complex for emulsion stabilization.

**Figure 3 foods-10-01354-f003:**
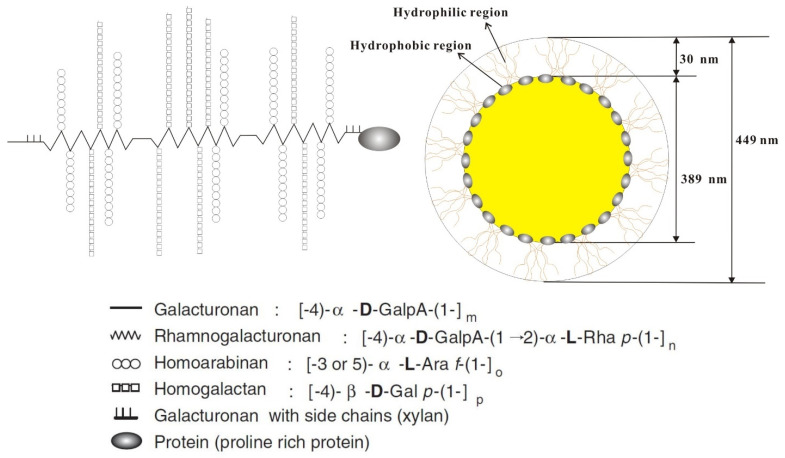
The chemical structure of soybean soluble polysaccharide (SPSS) (**left**) and a schematic diagram of SSPS-stabilized emulsion droplets (**right**).

**Figure 4 foods-10-01354-f004:**
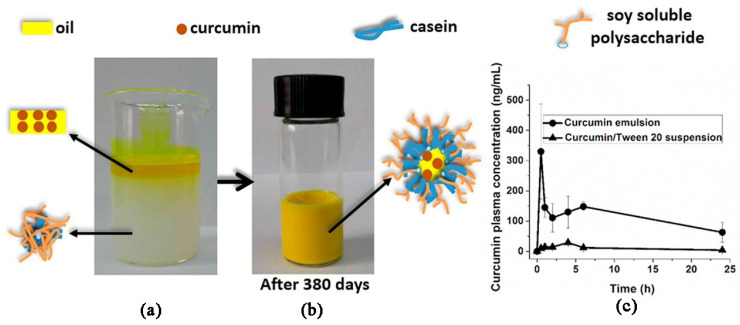
Diagrammatic representation of formation of SSPS-casein complex (**a**), the stabilized emulsion with a long-term stability (**b**), and the significantly improved loading and bioavailability of SSPS-casein stabilized curcumin emulsion compared to that stabilized by Tween 20 (**c**) [56].

**Figure 5 foods-10-01354-f005:**
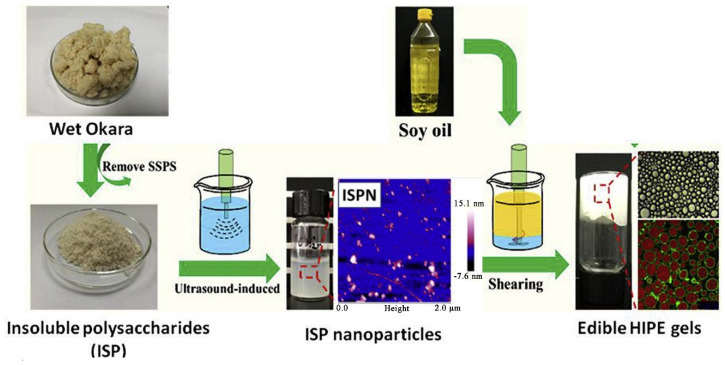
Synthesis of ultrasound-induced insoluble soy polysaccharide (ISP) based nanoparticles for developing and edible O/W high internal phase Pickering emulsion gels [60].

**Figure 6 foods-10-01354-f006:**
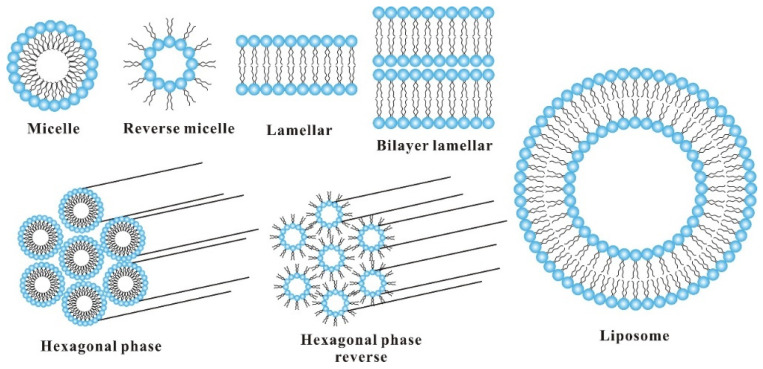
Diagrammatic presentation of various structures resulting from the self-assembly of phospholipids derived from soy lecithin.

**Figure 7 foods-10-01354-f007:**
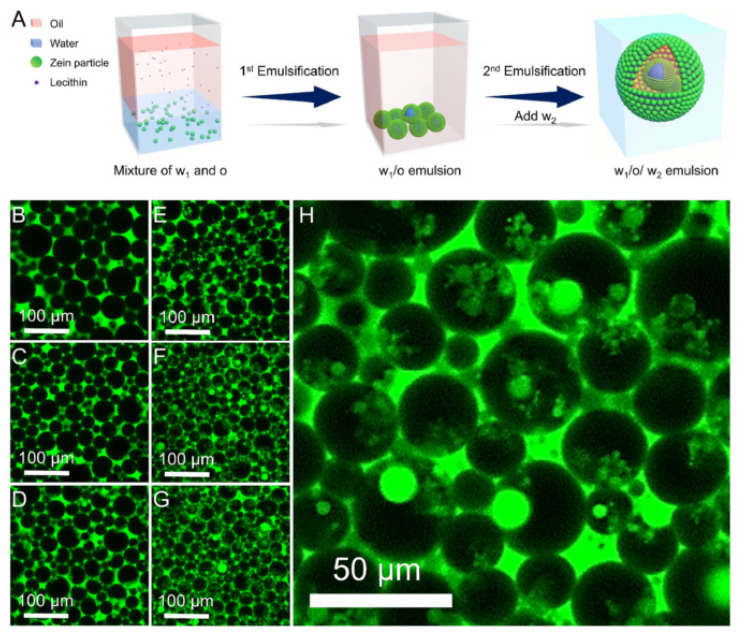
(**A**) Illustration of a two-step process for the preparation of lecithin based w/o/w high internal phase Pickering emulsions; (**B**–**G**) CLSM images of the emulsions stabilized with zein nanoparticles and lecithin, and the lecithin concentrations are 0%, 0.1%, 0.25%, 0.5%, 1%, and 2%, respectively; (**H**) CLSM image of the emulsion with 1% lecithin (**F**) at high magnification [69].

**Figure 8 foods-10-01354-f008:**
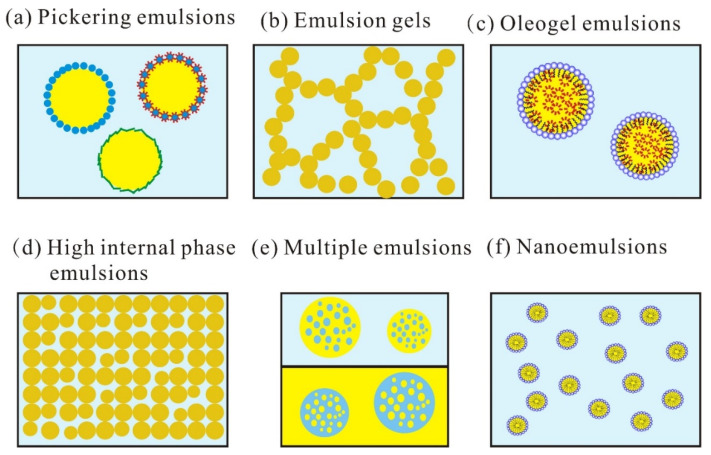
Structures of Pickering emulsions (**a**), emulsion gels (**b**), oleogel emulsions (**c**), high internal phase emulsions (**d**), multiple emulsions (**e**), and nanoemulsions (**f**).

**Table 1 foods-10-01354-t001:** Lists of recently reported studies focusing on polysaccharide glycation with soy proteins for improved emulsifying properties.

Emulsifier	Glycation Conditions	Main Conclusions	References
Soy protein isolate–glucose	Wet heating, 50, 60, 70, 80, and 90 °C, 5 h	The EAI and ESI of the soy protein–glucose isolate were markedly improved under different reaction temperature conditions in comparison to that of untreated SPI.	[37]
Soy protein–maltose	Wet heating, 100 °C, 2 h	The 1-butyl-3-methylimidazolium chloride was proved to be a proper medium for protein glycation to increase glycation extent and to improve the emulsifying activity and emulsion stability.	[38]
Soy protein isolate–gum acacia	Dry heating, 60 °C, RH 79%, 6 days	The soy protein isolates–gum acacia (SPI-GA) conjugates films containing essential oils showed the highest radical scavenging activity and antibacterial activity.	[39]
Soy protein hydrolysate-dextran	Wet heating, 85 °C, 1 h	The soy protein hydrolysate-dextran conjugates produced through the wet method under optimal conditions showed the lowest creaming index and the best freeze–thaw stability.	[36]
Soy protein isolate–Okara dietary fibre (ODF)	Dry heating, 60 °C, RH 78%, 6–72 h	The resulting ODF-SPI conjugates were thermally stable and exhibited excellent Pickering emulsion stabilization potentials.	[40]
Soy glycinin–soy polysaccharide	Dry heating, 60 °C, RH 78%, 24 h and 72 h	The glycation with soy soluble polysaccharide (SSPS) greatly improved the emulsification performance of soy glycinin, the gel network formation and stability (against heating or freeze–thawing) of the resultant high internal phase emulsions.	[41]
Soy protein–pectin	Wet heating, pH 4.5, 95 °C, 30 min	With the addition of glycyrrhizic acid nanofibrils, self-standing soy protein-pectin nanoparticles (SPNPs) stabilized emulsion gels with small droplet size, homogeneous appearance, and microstructure were obtained.	[42]
Soy protein isolate–pectin	Dry heating, 60 °C, RH 79%, 1–7 days	The solubility and emulsifying properties were improved after the Maillard reaction and the strong steric-hindrance effect of pectin facilitated the stability of the emulsion.	[43]

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
