# Peer review of "Current Progress in the Utilization of Soy-Based Emulsifiers in Food Applications—A Review"

_foods, 2021, doi:10.3390/foods10061354_

Round 1

Reviewer 1 Report

I had gone in detail on an earlier submission of this paper that is attached, and the latest changes do not alter my concerns, not specifically listed in the attached review. Dr. Lingli Deng has made a good effort to try to summarize the literature. The biggest problem is the literature itself. Although the measured property data are likely to be accurate, the soy material used and interpretation of the mechanism behind the data are both very problematic. Commercial soy protein isolate has changed over the year and there are very dramatic differences in the products available; thus the measured performance properties are dependent upon the soy source. The fact that the soy material is altered through enzymatic, thermal, and jet cooking processes1-3 makes knowing what soy material that was used is in this paper and the general literature glossed over. Additionally, the native SPI is a very different animal than commercial SPI.2 The author should invoke this caveat about the soy source in the beginning of the paper.

I get frustrated at the soy literature in that speculation on mechanisms for changes in properties are treated as sound information. This concern exists on the change in soy surface polarity, specific interactions, and especially about the very complex chemistry of the Maillard reaction. This “reaction” is provided as a general answer even though data is not presented that occurs under milder conditions needed for it to take place and the multitude of products are treated as a singular change in soy properties. Another issue is that the ultrasonic cavitation process is not considered as a thermal process even though the individual cavitations reach thousands of degree in temperature.

I am approving this paper with the author going over the paper to clearly indicate what is known data and proposed mechanisms being consistent with a certain hypothesis. One needs to clearly define what is fact and what is speculation. The author needs to do this since her is the one familiar the literature being cited.

The illustrations are hard to follow since the captions and text body do not discuss well what is illustrated. The author should modify the drawing to only what is being discussed in relation to soy solution structure or expand the text to explain what the reader should be taking away from the Figures.

                (1)          Egbert, W. R. In Soybeans as Functional Foods and Ingredients; Liu, K., Ed.; AOCS Press: Champaign, IL, 2004, p 134.

                (2)          Lorenz, L.; Birkeland, M.; Daurio, C.; Frihart, C. R. Forest Products Journal 2015, 65, 26.

                (3)          Wang, C.; Johnson, L. A. Journal of the American Oil Chemists' Society 2001, 78, 189.

Reviewer 2 Report

This is an interesting article about the potential soy based emulsifiers in food application. 

It explains the challenges and future potential of soy-derived emulsion in the food industry.

It is a relevant and interesting topic and is well written.

However, I would like to recommend the following revision in the manuscript.

1) There is still a very high similarity with the previously published articles. 

Author should minimize the similarity with previously published articles. 

2) Font sizes in Figure 2 and 3 are too small. They should be made more clear.

3) Figures 4 and 5 seem to be from other sources. It should be made clear whether permission was obtained from them or the author redraws them based on the ideas in these papers. 

4) Same for Figure 7.

5) I suggest adding a new section about the challenges of soy based emulsifiers and how they can be overcome.   6) please explain in detail about the advantages of soy-based emulsifiers over other plant-based emulsifiers.

Round 2

Reviewer 1 Report

Thanks for taking seriously my suggestions.

Please use the clarified version below for the summary

Commercial soy protein isolate has changed over the years and there are very dramatic differences in the products available, thus the measured performance properties are dependent upon the soy source. The fact that the soy material has been altered through enzymatic, thermal, and jet cooking processes [88-90] makes the comparison of soy ingredients used in literature more difficult.

This manuscript is a resubmission of an earlier submission. The following is a list of the peer review reports and author responses from that submission.

Round 1

Reviewer 1 Report

This review article discussed the application of soy-based emulsifiers in food industry. The effect of modification process on the emulsifying properties of soy protein and expanding its food application as emulsifier as well as soy polysaccharides and lecithin were reviewed. It is an interesting work and in line with the trend of plant-based ingredients for designing plant-based alternatives to animal-based products. It is a well-written manuscript. However, there remained some issues that need to be addressed. Consequently, the manuscript is recommended for a major revision.

-line 31-36: These sentences are in contrast with each other and they are vague. What do you mean by effective and efficient exactly?

-line 82: "as" is needed after considered and before an.

-Describe why modification is required for soy protein before mentioning different modification methods.

-line 132: Please mention the reason why ultrasound treatment decreased the foam stability of soy protein isolate?

-Definitely there are much more modification approaches which have been used for soy protein such as some chemical methods, some novel physical methods (thermal and non-thermal) and some biological methods such as fermentation. Why the author did not mention these methods?

- Providing a critical discussion comparing different methodologies (apart from than citing works that directly compare different methodologies themselves) would enhance the value of the manuscript.

- The conclusions (Future trends) do not add significant information to that provided in the introduction - again a critical discussion is missing.

Reviewer 2 Report

This is an interesting review in an area that has not been reviewed much recently. On a positive note, the article is well organized and clearly written. The 76 citations make it a good reference source.

However, this is not a critical review in that it takes all the literature at face value, but a lot of the literature is incorrect since it does not agree with many aspects of fundamental protein properties. Two significant areas of misinformation are due to denaturation and the Maillard reaction. The author basically took the cited publications as being totally true. A good review would question the logic of some of these papers. I also found it surprising that there was no references to the pertinent chapters in S. Damodaran, A. Paraf: Food Proteins and Their Applications, (Marcel Dekker, Inc., New York 1997).

Denaturation is a very complicated subject, and the complicated aspects are usually brushed aside with simple rational. An example on line 89 is the statement “increased molecular flexibility because of increased a-helix and random coil”. How does an increased α-helix content lead to flexibility, when it is a rigid structure? The low energy required to change protein states should make most proteins mobile in water when they are not in a crystalline state. How does one measure protein mobility under a specific condition since the external environment is part of the overall energetics of a protein structure. The statement that denaturation makes a less hydrophobic state, which is certainly logical when adsorbing onto air bubbles, but they would be less stable in aqueous environment leading to aggregation to reduce surface energetics. Native soy flour in water is quite good in foaming under stirring conditions.

On line 180, the statement is made that glycation, otherwise known as Maillard reaction endows food proteins with improved functional properties, such 140 as emulsifying properties, and it occurs under mild and safe conditions and requires no extraneous chemicals. In the first glycation is an entirely different process than the “Maillard reaction”, which has a whole complication chemical process. Certainly these two reactions involve reacting carbohydrates with protein, but the chemistry and products are completely different. The Maillard reaction often invoked in soy literature, but there is little evidence that occurs under most of the conditions used in processing soy. Certainly the reactions take place in the high temperature of baking, but not under the mild aqueous conditions used in the discussed soy modifications.